# Current Knowledge of IL-6 Cytokine Family Members in Acute and Chronic Kidney Disease

**DOI:** 10.3390/biomedicines7010019

**Published:** 2019-03-13

**Authors:** Aaron L. Magno, Lakshini Y. Herat, Revathy Carnagarin, Markus P. Schlaich, Vance B. Matthews

**Affiliations:** 1Research Centre, Royal Perth Hospital, Perth, WA 6000, Australia; aaron.magno@uwa.edu.au; 2Dobney Hypertension Centre, School of Biomedical Science—Royal Perth Hospital Unit, University of Western Australia, Crawley, WA 6000, Australia; lakshini.weerasekera@uwa.edu.au; 3Dobney Hypertension Centre, School of Medicine—Royal Perth Hospital Unit, University of Western Australia, Crawley, WA 6000, Australia; revathy.carnagarin@uwa.edu.au (R.C.); markus.schlaich@uwa.edu.au (M.P.S.); 4Department of Cardiology and Department of Nephrology, Royal Perth Hospital, Perth, WA 6000, Australia

**Keywords:** IL-6, kidney disease, cytokines, pre-clinical studies

## Abstract

Healthy kidneys are important for the efficient regulation of metabolism. However, there is an ever increasing population of patients suffering from both acute and chronic kidney diseases that disrupt this homeostasis. This review will explore the emerging roles that interleukin 6 (IL-6) cytokine family members play in the pathogenesis of kidney disease. The IL-6 family of cytokines are involved in a diverse range of physiological functions. In relation to kidney disease, their involvement is no less diverse. Evidence from both preclinical and clinical sources show that IL-6 cytokine family members can play either a deleterious or protective role in response to kidney disease. This appears to be dependent on the type of kidney disease in question or the specific cytokine. Current attempts to use or target IL-6 cytokine family members as therapies of kidney diseases will be highlighted throughout this review. Finally, the involvement of IL-6 cytokine family members in kidney disease will be presented in the context of three regularly overlapping conditions: obesity, hypertension and diabetes.

## 1. Introduction

As a pleiotropic cytokine, interleukin 6 (IL-6) is involved in a broad spectrum of biological events including, but not limited to (i) inflammation, which is a response to infections and damage in which the host defense eliminates the offending agents; (ii) glucose metabolism, which is the production and utilisation of glucose and (iii) haematopoiesis, which is the formation of blood cells [1]. All of these biological events are in some way shaped by the influence of IL-6 family cytokine signalling. Aside from IL-6, members of the IL-6 cytokine family include interleukin-11 (IL-11), leukaemia inhibitory factor (LIF), oncostatin M (OSM), ciliary neurotrophic factor (CNTF), cardiotrophin-1 (CT-1), cardiotrophin-like cytokine factor (CLCF-1) and granulocyte colony-stimulating factor (G-CSF) [2]. While there is overlap in the functions that IL-6 and its family members participate in, they are also involved in a diverse range of physiological (natural state) and pathophysiological (disease state) functions that are distinct from each other [3]. For example, the physiological process of osteoclast formation is one that is promoted by a number of IL-6 cytokine family members, including IL-6, IL-11, OSM and CT-1 [3]. On the other hand, a pathophysiological event such as the activation of T helper 17 cell populations (involved in the clearance of pathogens at mucosal surfaces) is promoted by IL-6 and IL-11, but inhibited by OSM activity [4,5].

IL-6 cytokine family members have been found to be elevated in the renal tissue of patients with kidney diseases, including diabetic nephropathy, glomerulonephritis and obstructive nephropathy. Cells of the kidney that express and secrete IL-6 cytokine family members include podocytes, endothelial cells, mesangial cells and tubular epithelial cells. In these cell types, IL-6 cytokine family member signalling can promote cell proliferation, impact differentiation or increase tubulointerstitial fibrosis [3]. In this review, we discuss the evidence from patient and preclinical models of kidney disease that shows the involvement of IL-6 cytokine family members in the pathology of these renal conditions. We also explore the possibilities of targeting IL-6 cytokine family signalling as viable therapies for the treatment of a range of kidney diseases, as elevated levels IL-6 cytokine family members is a common feature of these diseases. Therapeutic strategies being examined include monoclonal antibodies and fusion proteins to inhibit IL-6 cytokine family signalling.

## 2. IL-6 Cytokine Family Members

The IL-6 cytokine family is a group of cytokines that possess a similar four-helical bundle structure and share a common signalling subunit in glycoprotein (gp130) [6]. The sharing of the gp130 receptor allows for some redundancy between the cytokine family members [7]. While gp130 is expressed on all cells, the receptors for the individual IL-6 cytokine family members are cell specific, which restricts the cell types that respond to these cytokines, as gp130 alone is unable to bind the cytokines [6]. Renal tissue cells may express specific IL-6 cytokine family member receptors but in those that do not, IL-6 cytokine family signalling relies on the ubiquitously expressed gp130 beta receptor and a combination of the soluble alpha receptors. Signalling pathways induced by IL-6 cytokine family members include signal transducers and activators of transcription (STAT), mitogen-activated protein kinase (MAPK) and phosphoinositide 3-kinase (PI3K) [8]. The IL-6 cytokine family members exhibit a multitude of both beneficial and pathogenic consequences; hence, why they may be referred to as a “double-edged sword” [9]. Quite often the pathogenic outcomes result due to the signalling surpassing a critical threshold [10]. For example, IL-6 expression increases with the severity of an injury to the kidney, promoting the damaging inflammatory response, but also protecting the kidney from further acute injury by acting via its soluble receptor [10]. The following is a list of IL-6 cytokine family members and their respective binding partners.

### 2.1. Interleukin-6

Originally identified in 1986, IL-6 target cells include lymphocytes, myeloid cells, epithelial cells and hepatocytes, to mention a few [4,11]. IL-6 binds to dimers of the IL-6 receptor (IL-6R) and gp130, as well as a soluble version of the IL-6R (Figure 1). IL-6 binding to the IL-6R/gp130 dimer is considered classic signalling, which results in an anti-inflammatory response [12]. Trans-signalling is when IL-6 interacts with the soluble IL-6R/gp130 dimer and initiates a pro-inflammatory response [12]. Soluble IL-6R is generated by alternative splicing, proteolytic release of the ectodomain of the membrane-bound IL-6R or through shedding, which can be triggered by a multitude of factors including the depletion of cellular cholesterol [13]. Therapies, such as the sgp130Fc protein, have been developed that target soluble IL-6R trans-signalling and may be beneficial in treating various kidney diseases, which will be discussed in more detail later [14].

### 2.2. Interleukin-11

IL-11 was first discovered being expressed in an immortalized primate bone marrow-derived stromal cell line and subsequently identified in lung fibroblasts and chondrocytes where it plays a role in haematopoiesis, adipogenesis, neuronal differentiation and bone metabolism [15,16]. Elevated levels of IL-11 have been observed in a variety of cancers, including primary ovarian carcinoma, prostate carcinoma, breast cancer and colorectal carcinoma, where it is believed to drive cell proliferation, invasiveness and metastatic potential [16]. IL-11 binds to dimers of either IL-11Rα and gp130 or IL-11Rβ and gp130 [2].

### 2.3. Leukaemia Inhibitory Factor

LIF was independently cloned by multiple groups and is predominantly expressed in T cells, activated monocytes, fibroblasts and umbilical cord vein endothelial cells [17]. Studies have identified numerous biological roles for LIF, including a vital role in reproduction, bone remodelling, regeneration of neuronal and skeletal muscle cells in response to injury and protection of cardiomyocytes from injury [18]. LIF signals via a dimer of LIFR and gp130 [6].

### 2.4. Oncostatin M

OSM was initially discovered in the conditioned medium of histiocytic lymphoma cells that had been treated with phorbol 12-myristate 13-acetate [19]. Expression of OSM has been detected in monocytes, macrophages, neutrophils and T cells [20]. OSM has been found to act on hematopoietic cells, hepatocytes, cardiomyocytes and neurons [2]. Biological functions that have been attributed to OSM activity include haematopoiesis, bone turnover, lipid storage modulation, central nervous system development and liver regeneration [21]. OSM is able to signal through a dimer of either OSMR and gp130 or LIFR and gp130 [21].

### 2.5. Ciliary Neurotrophic Factor

Human CNTF was cloned from a human genomic library and mainly targets neurons and skeletal muscle cells, promoting their growth and regeneration [2,22]. Immunohistochemistry has identified CNTF expression in osteoblasts, osteocytes, osteoclasts and chondrocytes, where it is involved in bone metabolism [23]. The systemic administration of CNTF has been trialled as a treatment to promote weight-loss as CNTF has been shown to target multiple cell types involved in metabolism [24]. CNTF can affect satiety by acting on the hypothalamus, increasing insulin sensitivity in adipocytes and regulating glucose uptake in skeletal muscle [23]. CNTF primarily binds to a dimer of CNTFR and gp130. However, it is also capable of binding to a complex of IL-6R, gp130 and LIFR, although with a lower affinity [6].

### 2.6. Cardiotrophin-1

CT-1 was first identified in a model of cardiac hypertrophy and is predominantly expressed in cardiac myocytes, where it protects against apoptosis and induces cardiac hypertrophy [25,26]. Roles for CT-1 in the development and protection of nervous tissue, as well as the regulation of body weight and intermediate metabolism, have also been identified [25]. Currently, CT-1 does not have a unique receptor associated with it and binds to the dimer of LIFR and gp130 [2].

### 2.7. Cardiotrophin-Like Cytokine Factor-1

In 1999, CLCF-1 was discovered by two distinct methods, in a cDNA library screen of activated Jurkat human T-cell lymphoma cells and computationally from a large EST database [27]. Based on current evidence, including expression patterns, it is believed that CLCF-1 is involved in development; in particular, the development of the autonomic nervous system [28]. Like CT-1, it does not have a unique receptor and instead interacts with either dimers of LIFR and gp130 or CNTFR and gp130 [6]. It can also be secreted and form a complex with soluble CNTFR or cytokine receptor-like factor-1 (CRLF-1) [27].

### 2.8. Granulocyte Colony-Stimulating Factor

Human G-CSF was originally cloned from a cDNA library derived from a tumour cell line in 1986 by Souza et al., and is mostly found in macrophages and dendritic cells [2,29]. At the time, IL-6 was first identified, G-CSF was the only sequenced protein that shared significant homology with IL-6 with strikingly similar gene organisations [11]. While G-CSF is not always included as a family member, this IL-6 relative shares similar disulphide structures with IL-6 and forms a hexameric receptor complex like IL-6 [30,31]. While the primary role of G-CSF is the regulation of granulopoiesis, mutations of G-CSF have been associated with neutropenia and related diseases [32]. G-CSF binds to both colony stimulating factor 3 receptor (CSF3R) and gp130 [2].

## 3. Implications of IL-6 Cytokine Family Members in Kidney Disease

Kidneys are involved in a number of key regulatory functions that help to maintain whole body homeostasis. They filter blood, maintain fluid homeostasis and are crucial in the process of removing toxins through urine. Deteriorating renal function and chronic kidney disease are closely associated with cardiovascular morbidity and mortality [33]. Kidney diseases are commonly divided into two major groups based on the duration of the disease, i.e. acute kidney injury (AKI) and chronic kidney disease (CKD). IL-6 cytokine family members have been implicated in both AKI and CKD (Figure 2).

### 3.1. Acute Kidney Injury

The current definition for AKI was first established in 2004 with a standardized set of criteria [34]. The primary dimension assessed is the magnitude of serum creatinine elevation (change from baseline to peak creatinine), with an increasing emphasis on the duration of the AKI [35]. Based on this definition, the global incidence rate of acute kidney injury is 23.2% [33]. Importantly, high levels of circulating IL-6 in patients with acute kidney injury are predictive of an increased mortality rate [36].

#### Renal Ischemia and Reperfusion Injury

Patients undergoing major surgery (cardiac, liver, vascular or kidney) frequently suffer renal ischemia and reperfusion (I/R) injury as a serious complication. Renal I/R occurs when there is an initial restriction of blood supply to kidneys, which is followed by restoration of perfusion and concomitant reoxygenation [37].

Although CNTF is primarily expressed in neuronal tissue, the kidney is the next most abundant source of the cytokine. CNTF expression was found to be further increased in rat kidneys that had experienced an I/R renal injury [38]. The increase in CNTF levels paralleled the recovery of the renal structure following I/R suggesting it also has renal protective characteristics [38]. CNTF levels return to basal within 28 days of the I/R renal injury [38]. As LIF has a role in nephrogenesis, it was hypothesised that LIF could participate in renal regeneration following acute kidney injury [39]. In the days following an I/R event, LIF mRNA expression was significantly increased [39]. Yoshino et al., were able to demonstrate in both in vivo and in vitro models of I/R injury that LIF was specifically involved in the regeneration of damaged kidney [39]. As IL-11 had been shown to be cytoprotective in non-kidney tissues, a murine model of I/R kidney injury was used to examine the effectiveness of the IL-6 cytokine family member as a therapy for I/R injury [40]. The administration of human recombinant or PEGylated IL-11 before or after renal I/R reduced the level of renal necrosis, tubular injury, renal apoptosis and neutrophil infiltration [40]. Experiments in human kidney proximal tubule cells demonstrated that the renal protective characteristics of IL-11 were due to its ability to increase the nuclear translocation of HIF1-α and subsequently induce sphingosine kinase-1 (SK1) expression [40]. The induction of IL-11 has also been shown to be a key intermediary step in the renal protective effects of A_1_ adenosine receptor agonists, which also increases the expression of SK1 [41]. In a model of I/R injury, rats receiving CT-1 had better renal function and lower tubular damage than control rats [42]. However, there is still conflicting evidence from other rat models of renal injury on whether CT-1 treatment is protective [43] or disrupts renal function and causes renal damage [44]. These contrary results may be a dose-dependent effect or vary depending on the model of kidney injury. However, overall IL-6 family cytokines appear to be mostly protective in the context of renal I/R injury. As much of the research to date has focused on examining the effects of a single IL-6 cytokine family member at a time, experiments involving the combination of CNTF, LIF, IL-11 and CT-1 (a cocktail of IL-6 cytokine family members) for the treatment for I/R injury may prove highly beneficial.

### 3.2. Chronic Kidney Disease

CKD is a common disorder that is defined by structural or functional abnormalities of the kidney and/or a sustained reduction in the glomerular filtration rate. As a high risk condition, the global rise in the prevalence of CKD is alarming [45]. Between 1990 and 2010 chronic kidney disease rose in the rankings from twenty-seventh to eighteenth in the list of causes of total number of global deaths, which was the second largest rise up the list [46].

#### 3.2.1. Diabetic Nephropathy

The leading cause of CKD has become diabetic nephropathy (DN), which is morphologically defined by tubulointerstitial fibrosis and glomerular sclerosis [47]. DN is clinically categorised by progressive albuminaria and a decline in glomerular filtration rate [48]. Nearly 40% of diabetes patients develop DN, irrespective of their management of blood glucose and/or blood pressure [49,50]. Diabetic patients with DN were found to have elevated levels of IL-6 in their serum compared to diabetic patients without DN [51]. The likelihood of a diabetic patient to develop DN appears to be linked to polymorphisms in the IL-6 gene [47]. Following the observation that both IL-6 and soluble IL-6R were elevated in the sera of a DN patient population, Lei et al., examined the roles of classic and trans-signalling in a cell culture model. Interestingly, both classic and trans IL-6 signalling were responsible for renal damage [52]. Elevated levels of another IL-6 cytokine family member, OSM, were found in tubular epithelial cells from diabetic mice compared to those from nondiabetic mice [53]. There was also a higher level of tubulointerstitial fibrosis in the diabetic mice. Both the level of OSM expression and tubulointerstitial fibrosis were reduced in the diabetic mice via the overexpression of either SOCS1 or SOCS3 [53]. However, Sarkozi et al. has demonstrated that in human kidney proximal tubule cells, OSM stimulation can provide a protective effect against tubulointerstitial fibrosis [54]. This may be due to differences in the models used.

Therapies for DN that target IL-6 cytokine family signalling are at various stages of development. Blockage of IL-6 signalling using monoclonal antibodies targeting either IL-6 (Siltuximab) or its receptor IL-6R (Tocilizumab) are currently in clinical trials [47]. Signalling molecules downstream of IL-6 cytokine family receptors are also being targeted. There is a phase 2 randomised control trial being conducted using the Janus kinase inhibitor, bacitrinib, to treat DN in Type 2 Diabetes (T2D) patients by inhibiting IL-6-mediated JAK/STAT signalling [55]. The administration of a SOCS1 peptidomimetic to diabetic mice was shown to reduce the renal changes associated with DN, although its impact on OSM expression has not been measured [56]. Alternatively, administration of G-CSF is being examined as a possible therapy in a DN mouse model with evidence that it can reduce the level of renal damage [57].

#### 3.2.2. Glomerulonephritis

Glomerulonephritis exists as a range of kidney diseases that are characterised by damage to glomeruli. Diseases that fall under the umbrella of glomerulonephritis include IgA nephropathy and lupus nephropathy, amongst others.

In 1989, it was reported that transgenic mice expressing human IL-6 displayed the pathology of glomerulonephritis, including the profound cell proliferation of mesangial cells in the kidneys [58]. Subsequent studies examining IL-6 excretion in urine found higher levels in the urine of patients with glomerulonephritis compared to urine from healthy individuals or patients with other types of kidney disease. In fact, IL-6 urine levels increased with the severity of glomerulonephritis [59]. Evidence from a mouse model of lupus nephritis suggests that the increase in IL-6 is the result of a decrease in expression of a micro-RNA that regulates IL-6 [60]. This finding is supported by a study looking at the urine excretion of another IL-6 cytokine family member in glomerulonephritis patients [61]. Increasing levels of IL-11 excretion in the urine of IgA nephropathy and lupus nephropathy patients correlated with increased severity of their proteinuria [61]. This correlation was not observed in patients with proteinuria due to idiopathic nephrotic syndrome, indicating that this correlation is specific to glomerulonephritis [61]. Idiopathic nephrotic syndrome has an undetermined pathogenesis, shares the proteinuria symptoms observed in glomerulonephritis but renal biopsies show minimal renal changes unlike glomerulonephritis [62]. While examining the effect of LIF on the lymphoid system, Shen et al., generated transgenic mice that overexpressed LIF specifically in their T lymphocytes [63]. Among the pathologies found in these transgenic mice was a proliferative mesangial glomerulonephropathy with extensive hyaline deposits [63]. When mesangial or epithelial cells from the biopsies of patients with glomerulonephritis were cultured, they were found to express LIF at a higher level than those from non-diseased kidneys [64]. In a mouse model of lupus nephritis, elevated levels of OSM were detected in renal tissue [65]. In an effort to understand the mechanisms by which the elevation of IL-6 cytokine family members may cause renal damage, an angiotensin II infusion mouse model of hypertension and CKD was used [66]. Ablation of IL-6 in these mice attenuated the angiotensin II-induced hypertension and features of CKD, including proteinuria and renal fibrosis [66].

Targeting IL-6 signalling is being examined as a treatment for glomerulonephritis. A clinical trial using a neutralising monoclonal antibody to IL-6 to treat patients with lupus nephritis failed to show efficacy in ameliorating the features of the disease, including proteinuria [67]. A neutralising monoclonal antibody to the IL-6 receptor was found to preserve glomerular function and structure in a lupus nephritis mouse model, but failed to prevent the associated cell proliferation and proteinuria [68]. In a Lyn-deficient mouse, which has elevated IL-6 and lupus nephritis symptoms, treatment with the IL-6 trans signalling inhibitor sgp130Fc attenuated the disease and improved renal function [69]. In another study using a lupus nephritis mouse model, an anti-OSM antibody was shown to attenuate renal fibrosis and partially improve urinary protein excretion [65]. This finding is similar to what Liu et al. showed in an in vitro model of diabetic nephropathy, where less OSM correlated with less fibrosis [53]. Interestingly, studies using a rat model of glomerulonephritis demonstrated that administration of IL-11 could reduce the glomerular necrosis and proteinuria associated with the disease [70]. Further research using murine models of glomerulonephritis identified NF-Kappa B activity and TGF-β expression as key components involved in the reduction of renal injury due to IL-11 treatment [71,72]. G-CSF has been examined as a possible therapeutic agent for lupus nephropathy with mixed results. In the MRL-*lpr* lupus mouse model, a low dose of G-CSF was found to exacerbate the lupus nephropathy, while a high dose was able to prevent lupus nephritis [73]. A later study using the higher dose of G-CSF was examined in the NZB/W F1 lupus mouse model and corroborated the findings that G-CSF treatment could protect against lupus nephritis [74]. However, there are case reports of patients with systemic lupus erythematosus and associated glomerulonephritis that have received G-CSF treatment which have resulted in disease flares with a rapid and irreversible decline in renal function observed [75].

#### 3.2.3. Focal Segmental Glomerulosclerosis

Focal segmental glomerulosclerosis (FSGS) causes nephrotic syndrome and frequently leads to end-stage renal disease. It is diagnosed pathologically from renal biopsies and presents with varied clinical features and etiologies. Circulating permeability factors are believed to play a major role in the pathogenesis of FSGS with many studies examining the plasma of FSGS patients to identify these factors.

The IL-6 cytokine family member, CLCF-1, was found to be at levels 100 times higher in FSGS patient plasma compared to controls [76]. CLCF-1 was isolated from FSGS patient plasma via galactose affinity chromatography and identified by mass spectrometry [76]. An in vitro system was used to verify that CLCF-1 was a biologically active component involved in the pathogenesis of FSGS and was found to mimic the effects of FSGS patient plasma [77]. A monoclonal antibody was able to block the activity of FSGS patient plasma in this in vitro system [77]. The addition of the CLCF-1 heterodimer binding partner, CRLF-1, also attenuated the activity of FSGS patient plasma and CLCF-1 in the in vitro system [78]. As CLCF-1 was isolated by galactose affinity chromatography, galactose has been proposed as a treatment of FSGS, but so far this treatment has had limited success with only some patients showing a reduction in proteinuria with stable glomerular filtration rate [79]. As no essential role for CLCF-1 has been identified post-foetal development, antibodies targeting CLCF-1 or its receptors have been proposed as a potential treatment [77]. Further studies are warranted to elucidate the role CLCF-1 may have in homeostatic functions, as well as in diseases, both kidney related and in other organs. Recently, there has been a case report of a patient with FSGS associated with cutaneous and systemic plasmacytosis that had elevated IL-6 serum levels suggesting that other IL-6 cytokine family members may be involved in the pathogenesis of FSGS [80].

#### 3.2.4. Obstructive Nephropathy

Chronic obstructive nephropathy is a form of CKD that develops from the obstruction of the urinary tract. Tubulointerstitial fibrosis is a common morphological feature shared between obstructive nephropathy and other CKDs like DN. Elevated levels of OSM were detected in the renal tissue of patients that had developed obstructive nephropathy through variable means [81]. OSM was also found to be upregulated in a surgically created rat model of obstructive nephropathy [81]. Using the same model, Lee et al., showed that CNTF expression levels in their kidneys were well above those observed in the kidneys of sham-operated mice [82]. Unlike the short-term elevation of CNTF expression observed in renal I/R injury, CNTF expression remained elevated to day 28 in the obstructive nephropathy rat model [38,82].

## 4. Obesity, Hypertension, Diabetes and Kidney Disease

It has been reported that 70% of all cases of end-stage renal disease are related to central obesity, diabetes and/or hypertension [83]. As such, separating kidney disease from other conditions like obesity, diabetes and hypertension is difficult because they are all entwined. The global prevalence of obesity continues to increase with over 600 million adults and over 100 million children estimated to be obese worldwide [84]. Obesity causes a number of structural changes in the kidneys, including fewer nephrons and abnormal renal tubular exchange. This results in a decline in sodium excretion and an impaired diuretic response, which leads to an inability to effectively decrease blood pressure elevations [85]. Visceral adipose tissue in obese subjects can completely encapsulate the kidneys and could exert a compressive force on them which would increase the renal capsular pressure and subsequently increase arterial pressure [86]. Adipose tissue is also a major source of the metabolic factors known as adipokines, which provide the means of crosstalk between adipose tissue and the kidney. Under obese conditions there is an imbalance in this crosstalk that leads to renal damage [82]. We have shown that IL-6 appears to be protective against the renal changes that this imbalance of crosstalk in obesity creates. We observed that obese IL-6 knockout mice were far more susceptible to renal abnormalities than their obese wild type counterparts [87]. This result suggests that although IL-6 neutralising therapies have renal benefits, there is clearly a minimal level of IL-6 activity required for healthy renal function.

A high body mass index (BMI) contributed to approximately 7% of deaths globally in 2015 with cardiovascular disease and diabetes being the two leading causes [84]. While genetic predisposition is the underlying factor to developing T2D, obesity is also a driving force to the development of T2D, with endoplasmic reticulum stress being the molecular link between the two [88]. Briefly, the induction of ER stress signalling by obesity leads to reduced insulin receptor signalling, systemic insulin resistance and eventually T2D [88]. Alternatively, T2D leads to obesity as the inherent insulin resistance increases glucose production and insulin levels, resulting in obesity [89]. Diabetes can result in kidney diseases like the aforementioned DN, but also athero-embolic renal disease, ischemic nephropathy and interstitial fibrosis [90].

There is a high prevalence of hypertension amongst the obese population, with an increasing risk of developing hypertension being shown with weight gain [91]. The development of hypertension in obese patients can be dependent on a wide variety of factors, including an adipokine imbalance, insulin resistance, renal abnormalities, maladaptive immunity, gut microbiome alterations and activation of either the sympathetic nervous system (SNS) or the renin-angiotensin-aldosterone (RAAS) system [91]. As previously stated, renal damage associated with kidney disease can promote hypertension, as there is a decline in sodium excretion, which is essential for the regulation of blood pressure [85]. However, kidney disease can also progress the propensity for hypertension to increase the activity of the SNS or the RAAS system [85]. Activation of the SNS via afferent signals of sensory renal nerves is an early event in the pathophysiology of kidney disease [92]. Renal sympathetic activation leads to volume retention, stimulates the release of renin and the secretion of noradrenaline, the major neurotransmitter of the SNS [92]. As conditions such as obesity and T2D have hyperglycaemia occurring, we posed the question as to whether hyperactivation of the SNS, as evidenced by elevated noradrenaline signalling, may increase the primary protein involved in glucose reabsorption, sodium-glucose co-transporter 2 (SGLT2). We found that in human kidney proximal tubule cells, noradrenaline stimulation increased SGLT2 mRNA levels [93] and protein levels [94]. Of importance to the context of the current review, we have also demonstrated that IL-6 mRNA expression [93] and protein secretion were also elevated with noradrenaline stimulation [94]. This may be the mechanism behind the increase in SGLT2 as it has been previously documented that IL-6 directly increases SGLT2 expression [95]. SGLT2 expression has become increasingly relevant in terms of renal function as SGLT2 inhibitors continue to be trialled for the treatment of kidney disease [96].

## 5. Conclusions

As highlighted in this review, multiple members of the IL-6 cytokine family are involved in a variety of kidney diseases. Currently, levels and activity of IL-6 cytokine family members can only be used as surrogate markers for disease severity and progression [1]. Elucidating the mechanisms by which they act in kidney diseases is complicated, as in some instances they appear beneficial and in others pathogenic. Due to i) duality of roles; ii) redundancy between IL-6 cytokine family members and iii) the fact that kidney diseases often occur with other diseases like diabetes or obesity, further complicates their exact nature in specific diseases. To untangle this complicated interplay between cytokines and multiple diseases, requires further studies in vitro and in animal models. There are many therapies that target IL-6 cytokine family members to treat kidney diseases currently being investigated in clinical trials at various phases, which include kinase inhibitors and IL-6 neutralising antibodies. Efficacy of these strategies will depend on their ability to overcome the possible redundancy of multiple IL-6 cytokine family members impacting the same disease. Only with a greater understanding of the molecular mechanisms of IL-6 family cytokines and the interplay between them, can we develop better targeted therapies towards this promiscuous family of cytokines. With additional knowledge and more sophisticated therapies, targeting IL-6 cytokine family members will become a viable strategy to treat kidney diseases.

## Figures and Tables

**Figure 1 biomedicines-07-00019-f001:**
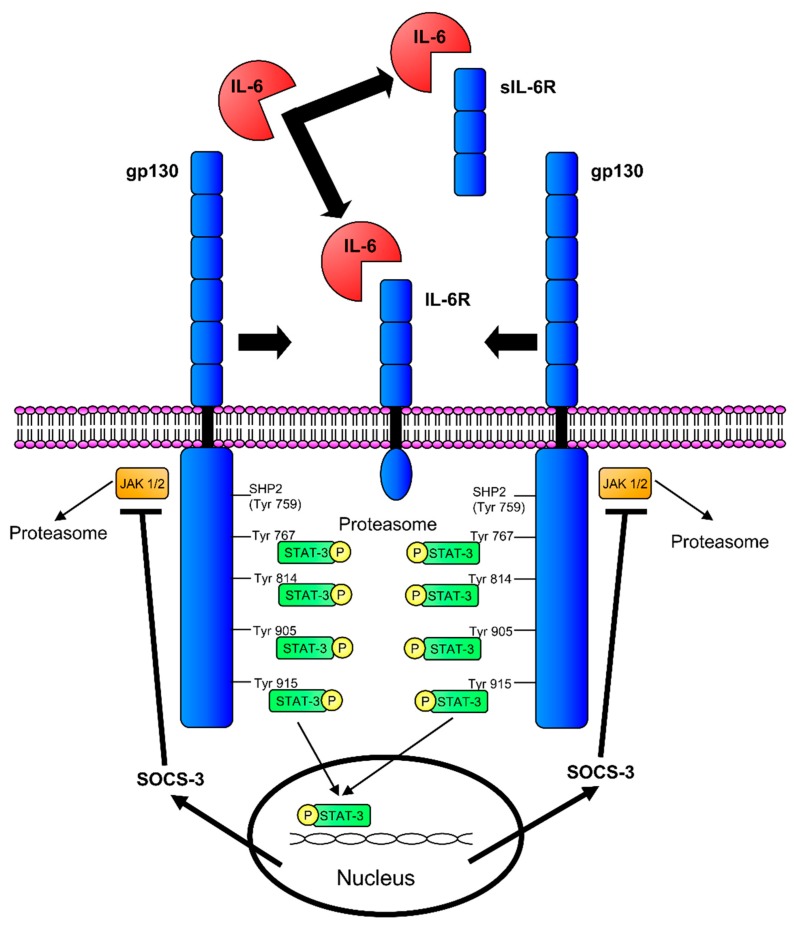
IL-6 signalling. IL-6 can signal by first binding to the membrane bound IL-6R or soluble IL-6R (sIL-6R). After recruitment of the gp130 receptors, Janus kinase (JAK)/STAT signalling occurs on the intracellular domains of the gp130 receptors. Suppressor of cytokine signalling-3 (SOCS-3) can inhibit JAK/STAT signalling on the gp130 receptors by binding the Src Homology Phosphatase 2 (SHP2)/Tyrosine (Tyr)^759^ binding sites.

**Figure 2 biomedicines-07-00019-f002:**
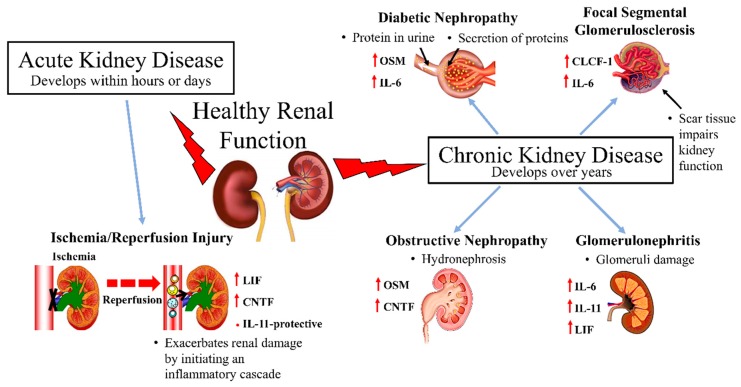
A diagrammatic overview of IL-6 cytokine family members’ relationship to various kidney diseases.

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
