# Peer review of "Current Knowledge of IL-6 Cytokine Family Members in Acute and Chronic Kidney Disease"

_biomedicines, 2019, doi:10.3390/biomedicines7010019_

Round 1
Reviewer 1 Report
Dear authors,
The manuscript entitled “Current Knowledge of IL-6 Cytokine Family Members in Acute and Chronic Kidney Disease” has been thoroughly reviewed. The subject proposed in the manuscript seems interesting and it provides relevant information on the topic. The paper is significantly accurate and fits in the scope of the journal. However, it need to be improved with other supporting information. There are few comments that should be clarified:
Introduction is insufficient and less informative. Please elaborate about inflammation, glucose metabolism and haematopoiesis that have been mentioned in the manuscript, and how these diseases are associated with IL-6 cytokine and its family? Line 33-35, Author should mention briefly about IL-6 family members. Author should provide example(s) for physiological and pathophysiological functions and how they are distinct from each other. Also, please discuss about the kidney diseases, and why IL-6 and its family cytokines are important to be targeted as therapy? Line 48-49, discuss about the harmful and beneficial consequences of IL-6 cytokine and its family members especially in association with kidney diseases. Which types of cells in Kidney secrete IL-6 and other member of its family and what are their effects? How IL-6 signals in Kidney? The conclusions also need to highlight what is new, the limitations, and how realistic are the expectations of IL-6 and its family being targeted in the future.
Author Response
We have addressed the following comments and concerns of the reviewer as follows;
1. Please elaborate about inflammation, glucose metabolism and haematopoiesis that have been mentioned in the manuscript, and how these diseases are associated with IL-6 cytokine and its family?
We have expanded on the introduction by elaborating on inflammation, glucose metabolism and haematopoiesis on Lines 31-34. We have also eluded to the role that IL-6 cytokines play in these mentioned diseases on Lines 82-83, Line 123, Lines 92-93 and Line 112.
2. Line 33-35, Author should mention briefly about IL-6 family members.
As suggested by the reviewer we have moved the listing of IL-6 cytokine family members from later in the manuscript to the introduction (Lines 36-38). The subsection entitled “2. IL-6 Cytokine Family Members” following directly after the introduction suitably contains the information the reviewer requested.
3. Author should provide example(s) for physiological and pathophysiological functions and how they are distinct from each other.
We have further highlighted the meanings of physiological and pathophysiological (Line 40) in the introduction and have provided examples of physiological/pathophysiological functions where there is overlap between the activity of IL-6 and its family members and where IL-6 cytokine family members act in distinct opposing manners.
4. Also, please discuss about the kidney diseases, and why IL-6 and its family cytokines are important to be targeted as therapy?
We have brought forward our first mention of the varied kidney diseases that are relevant to IL-6 family cytokines to the introduction (Lines 46-47). We want to highlight all of section 3 is dedicated to thoroughly discussing the relevant kidney diseases. The introduction now contains information about kidney disease therapies and why IL-6 family cytokine family members should be targeted. We want to reiterate that therapies are further discussed on Lines 177-184, 211-220, 252-273, 291-293.
5. Line 48-49, discuss about the harmful and beneficial consequences of IL-6 cytokine and its family members especially in association with kidney diseases.
We have briefly touched on the harmful and beneficial consequences of IL-6 in association with kidney diseases at Lines 71-73. Further examples of opposing actions of specific IL-6 cytokine family members in kidney disease appear later in the manuscript as originally presented. Lines 200-204: CT-1 in renal ischemia and reperfusion injury. Lines 227-233: OSM in diabetic nephropathy.
6. Which types of cells in Kidney secrete IL-6 and other member of its family and what are their effects?
We have included additional information about the types of kidney cells that secrete IL-6 cytokine family members and what their cellular effects are in the introduction (Lines 48-51).
7. How IL-6 signals in Kidney?
We have made an addition at Line 64-66 to indicate how IL-6 and other members of its family signal in the kidney.
8. The conclusions also need to highlight what is new, the limitations, and how realistic are the expectations of IL-6 and its family being targeted in the future.
There are clinical trials using drugs targeting IL-6 cytokine family members to treat kidney diseases currently ongoing, including many in Phase 2. It would be premature to provide a strong statement addressing the utility of these drugs at this stage. However, a statement has been added to the conclusion to elude to this and highlight one of the major limitations of the therapies, which is the redundancy that exists between IL-6 family cytokines (Lines 396-404). We are optimistic that IL-6 and its family members can be targeted in the future to treat kidney disease.
Reviewer 2 Report
This review highlights the role of IL-6 family members in distinct types of acute kidney injury and chronic kidney disease. Currently, IL-6 is recognised as a central cytokine in the pathogenesis of cardiovascular disorders, particularly including those involving ectopic calcification, and it also plays a major role in kidney diseses as stated in the review. This paper describes the biology of IL-6 family members, their signaling pathways, and pathophysiological importance. Despite the narrative rather than critical style of the review, it gives the reader a brief but concise understanding of this particular field, and can be recommended for the initial reading before going to more sophisticated and detailed review papers. The quality of writing is acceptable. Overall, I would consider this review as meriting the publication in its present form.
Author Response
We appreciate the comments of the reviewer who has recommended accepting the manuscript without changes, especially in understanding the intent of the review.